# Green Banana (*Musa acuminata AAA*) Wastes to Develop an Edible Film for Food Applications

**DOI:** 10.3390/polym13183183

**Published:** 2021-09-19

**Authors:** Diego Salazar, Mirari Arancibia, Santiago Casado, Andrés Viteri, María Elvira López-Caballero, María Pilar Montero

**Affiliations:** 1Facultad de Ciencia e Ingeniería en Alimentos, Universidad Técnica de Ambato, Av. Los Chasquis y Rio Payamino, Ambato 180206, Ecuador; marancibias@uta.edu.ec (M.A.); s.casado@uta.edu.ec (S.C.); andresviteri11@hotmail.com (A.V.); 2Facultad de Veterinaria, Universidad Complutense de Madrid, 28040 Madrid, Spain; 3Instituto de Ciencia y Tecnología de Alimentos y Nutrición (ICTAN-CSIC), Calle José Antonio Novais 10, 28040 Madrid, Spain; elvira.lopez@ictan.csic.es

**Keywords:** green banana flours, discarded banana, edible film, snack, grilled wrapped chicken, storage

## Abstract

In this study, edible packaging based on discarded green banana (*Musa acuminata AAA*) flour (whole banana and banana peel flours) was developed for food applications. Films were characterized in terms of film-forming ability, mechanical, barrier, thermal, microbiological, and sensory properties. The film forming solutions were studied for rheological properties. Two formulations were selected based on their film-forming ability: whole banana flour (2.5%), peel flour (1.5%) and glycerol (1.0 %, F-1.0 G or 1.5%, F-1.5 G). Adding 1.5% plasticizer, due to the hygroscopic effect, favored the water retention of the films, increasing the density, which also resulted in a decrease in lightness and transparency. Water activity shows no difference between the two formulations, which were water resistant for at least 25 h. DSC results showed a similar melting temperature (Tm) for both films, around 122 °C. Both films solutions showed a viscoelastic behavior in the frequency spectrum, being the elastic modulus greater in F-1.0 G film than F-1.5 G film at low frequency. F-1.0 G film was less firm, deformable and elastic, with a less compact structure and a rougher surface as confirmed by AFM, favoring a higher water vapor permeability with respect to F.1.5 G film. Microorganisms such as *Enterobacteria* and *Staphylococcus aureus* were not found in the films after a period of storage (1 year under ambient conditions). The F-1.0 G film with added spices (cumin, oregano, garlic, onion, pepper, and nutmeg) was tested for some food applications: as a snack (with or without heat treatment) and as a wrap for grilled chicken. The performance of the seasoned film during chilled storage of chicken breast was also studied. Sensory evaluation showed good overall acceptability of all applications. In addition, the chicken breast wrapped with the seasoned film registered lower counts (1-log cycle) than the control (covered with a polystyrene bag) and the film without spices. Green banana flour is a promising material to develop edible films for food applications.

## 1. Introduction

The agro-industrial sector generates a significant amount of discards, which can be used as recovered raw material to produce and add value to existing or new products [1]. Discards are also deposited in landfills or discarded in inadequate places, contributing to the increase in biomass production. The accumulation of biomass contributes to generating problems such as the proliferation of microorganisms, which on a large scale could be associated with the production of greenhouse gases, a proliferation of pathogenic bacteria and fungi, toxic degradation of products, also the increase in the amount of biomass are related with fires in regions with a drier climate [2]. However, in the discards, it is possible to find components that provide nutrients such as vitamins, minerals, fibers, carbohydrates, proteins, natural colorants, antimicrobial compounds, and antioxidant compounds; therefore, given the characteristics of these elements, they can be used in different food applications such as natural food ingredients, additives, and supplements with high added value [3].

Banana production is one of the most important crops worldwide; its production was estimated at 116 million tons worldwide in 2017–2019, with an approximate value of USD 31 billion [4]. However, this crop has great losses; postharvest losses for green bananas that are rejected for certain deficiencies and not being exportable fluctuate around 5% and 10% [5], constituting very low-value products or even waste. The primary use of these residues or discards is for animal feed or as fertilizer, although other high value uses for human consumption and food are currently being examined, such as obtaining banana flour, a stable material [6]. Recently, the characterization and techno-functional properties of flours from whole green banana or its pulp have shown their potential as functional ingredients due to their starch, protein, high-fiber, and low-fat content [6]. In addition, their low-sugar and high-starch (especially resistant starch) [7] content make green banana flour a compound of great interest for potential applications, reducing the environmental impact at the same time.

The development of edible packaging materials has been the subject of many studies in order to extend the shelf life of foods, microbiological control, improve sensory attributes, etc. Due to the interest in replacing plastic packaging in the food industry, especially single-use packaging, the search for new biodegradable materials is necessary [8]. In this respect, underutilized materials, residues, or waste from the agri-food industry acquire a relevant role as substitutes in generating new packaging [9]. Biopolymers from these natural sources are respectful of the environment, being renewable and biodegradable while eliminating waste or by-products, thus increasing their added value [10].

Starch is one of the most widely used biopolymers in the development of biodegradable plastics [11]. This biopolymer, which is especially cheap, can be obtained from by-products of the agri-food industry. Starch-based films are low-cost and confer rather good transparency; in fact, starch-based films account for more than 60% of all biodegradable films [9]. Starch sources are very varied, and while most of them come from cereals or even tubers, one of the most important fruit sources is bananas. 

Banana flours (from the pulp or the whole banana) have the ability to form filmogenic solutions [6]. In this context, banana peel flour is a by-product with high starch and fiber contents. The addition of banana peel flour in films decreases the film solubility [12]. Banana peel flour is used with mixtures of other compounds such as polyvinyl alcohol [13], commercially isolated corn starch [14], banana starch, and cellulose nanocrystals as reinforcements [15], and tapioca starch [16] for the development of films. These mixtures have been obtained in order to mitigate the limitations that each component could present concerning to the mechanical properties or water resistance [12,16]. However, to the best of our knowledge, there are no studies regarding edible films based on discarded whole green bananas and banana peels with any food applications.

The present work aims to obtain and characterize edible films based on whole banana flour using banana peel flour as reinforcement to make integral use of the fruit. In addition, a film with added spices was also applied in two different ways: as a snack-type product (with or without heat treatment) and as an edible packaging used to wrap chicken during chilled storage or grill.

## 2. Materials and Methods

### 2.1. Banana Flours

Green bananas (*Musa acuminata AAA*), which did not satisfy the minimum requirements for export, were purchased from crops in Valencia, Los Rios (Ecuador). Bananas were cleaned and washed with water. Two types of slices of ~3 mm were obtained as follows: the first type were whole banana slices (pulp and peel), and the second type consisted of slices from the peel (banana peel). All slices were dried at 60 °C for 6 h in a convective oven dryer (Gander MTN, Saint Paul, MN, USA) until constant moisture values. Finally, the dried product was milled to produce a fine powder. All samples were hermetically packed and stored at room temperature until further analysis. The composition of the resultant flours on a wet basis was as follows: whole banana flour (moisture 11.32%, fat 0.87%, protein 3.53%, ash 3.76%, fiber 3.51%, carbohydrates 77.03%), and peel banana flour (moisture 9.64%, fat 3.51%, protein 4.24%, ash 6.35%, fiber 10.26%, carbohydrates 66%).

### 2.2. Edible Film Preparation and Screening of Film-Forming Ability 

The edible film was developed by casting according to the methodology described by Arancibia, et al. [17]. The banana flours were dissolved in distilled water (100 mL) at room temperature until complete dissolution, then glycerol was added. The ratios of flour and glycerol are shown in Table 1. The mixture was heated at 90–95 °C, kept at this temperature for 5 min, and allowed to cool at 50 °C. The filmogenic solutions (25 g) were cast in Petri plates (9 cm diameter) and dried on a stove at 60 °C for 8 h. Then, the dried films were conditioned at 22–23 °C for 72 h, maintaining the relative humidity (RH) at 58%.

The film-forming ability and film transparency were first qualitatively determined to select the most proper film formulation. The film-forming ability was evaluated based on the capacity of the flour blends to uniformly cover the surface of the plate, forming a smooth film upon drying. According to this qualitative premise, the film-forming ability was arbitrarily classified as (−) no ability, (+) moderate, and (++) good. The ability to detach from the Petri plates was classified as does not peel off (−), does not peel off easily (+), peels off easily (++). Manageability and texture were classified as: brittle and discontinued structure (−), flexible, manageable, coarse, and rough surface (+), and flexible, manageable, consistent, and smooth surface (++). The visually transparency was classified as: very transparent (+++), transparent (++), translucent (+), opaque (−). 

Based on the qualitative evaluation, two filmogenic formulations were selected and characterized (F-1.0 G film and F-1.5 G film).

### 2.3. Rheological Properties of Filmogenic Solutions

The rheological tests were performed in a Compact Modular Rheometer (Anton Paar MCR302, Graz, Austria) using a cone-plate PP25 for oscillatory tests. A sample of the filmogenic solution (~1 mL) was placed in the rheometer and compressed to a gap size of 1 mm. A thin liquid paraffin layer was used onto the edge of the sample to prevent moisture loss. A strain of 1% during the frequency range of 1 to 100 rad/s was used. The data obtained, storage modulus (G′) and loss modulus (G″), were analyzed with the Rheo Compass software. Determinations were made at 20 °C in triplicate.

### 2.4. Moisture, Thickness, and Density of Edible Films

The moisture content of the films was analyzed gravimetrically, drying the samples at 105 °C for 24 h. The thickness was measured with a Fowler micrometer (with an accuracy of 0.0001 mm) from an average of 15 random measurements. The density of the films was developed by weighing 4 cm × 4 cm rectangles and performing at least 10 measurements.

### 2.5. Water Activity, Water Vapor Permeability, and Solubility of Edible Films

Water activity (aw) tests were carried out with the water activity meter (Aqualab 4TE, Sao Paulo, Brazil), previously calibrated using saturated salt solutions; according to the calibration procedure for this instrument, water activity was measured at a constant temperature (25 °C). Water vapor permeability (WVP) was determined following the method described by Sobral, et al. [18] at 21 °C in a desiccator with distilled water (100% relative humidity). The films were firmly fixed on the opening of the cells containing silica gel; these cells were weighed every hour for 3 days. The water vapor permeability was calculated with Equation (1).
(1)$$WVP=\frac{w}{tA}\;\frac{x}{\Delta P}$$
where the term *w*/t was calculated by linear regression from the points of weight gain and time in the constant rate period. *A* is the permeation area (cm^2^), *x* is the average thickness of the films, and Δ*P* is the difference of partial vapor pressure of the atmosphere with silica gel and pure water (2642 Pa a 22 °C).

The solubility was calculated as described by Arancibia, et al. [19]. Films were cut into a square form with 4.5 mm length and placed in containers with 30 mL distilled water at 22 °C for 24 h. The solution was filtered to recover the remaining undissolved film portions, which were desiccated at 105 °C for 24 h. Film solubility, expressed as Total Soluble Matter TSM (%), was calculated using Equation (2).
(2)$$Solubility=\frac{Wo-Wf}{Wo}*100$$
where *Wo* was the initial weight of the film expressed as dry matter and *Wf* was the weight of the undissolved desiccated film residue. All determinations were made in triplicate.

### 2.6. Color and Light Barrier Properties of Edible Films

The color of the films was measured by a colorimeter (ColorFlex EZ, HunterLab, Reston, VA, USA) using the CIELAB color scale with parameters L* (lightness), a* (red/green), b* (yellow/blue). Color determinations were made at least 15 times per sample. The transparency and UV light absorption capacity of the films were evaluated using a UV-Vis spectrophotometer (Multiskan GO, Thermo Scientific, Waltham, MA, USA). The percentage of transmittance was calculated at wavelengths 280 and 660 nm. The analyses were conducted at least in triplicate. 

### 2.7. Mechanical Properties of Edible Films

Tensile properties were determined by the standard method of the American Society for Testing and Materials (ASTM) D882 (ASTM, 2002). The films were cut into rectangles 84 mm long and 21 mm wide and were conditioned for analysis in a desiccator with a saturated NaBr solution for 24 h at 52% RH. The samples were placed in the clamping jaws of a Brookfield CT3 10 K texturometer. The initial clearance between the jaws was 32.5 mm, and a strain rate of 0.5 mm/s was used. Tensile strength (force/initial cross-sectional area) and elongation at break (EAB) were calculated directly from the stress vs. strain curves. The elastic modulus was calculated as the slope of the initial linear portion of this curve. The results were expressed as an average of at least ten repetitions. The film puncture test was determined using a Brookfield CT3 10 K texture analyzer; 100 mm × 100 mm rectangular films were used and attached to a base of two fixed plates with a 40 mm diameter internal circular aperture. A 12.7 mm diameter TA18 stainless steel probe was used, the tests were carried out with an activation load of 0.05 N and a speed of 0.2 mm/s. The puncture force (N) and puncture deformation (mm) were obtained using the Texture Pro CTV 1.2 Build 9 software [20]. All the determinations were carried out at least in triplicate.

### 2.8. Water Resistance of Edible Films

Water resistance was evaluated according to the method described by Arancibia, et al. [21]. The films were fixed onto the opening cells with an area of 16 cm. Then, 2.5 mL of distilled water was poured onto the surface of the film, and the deformation of the film (cm) as well as the volume of water that filtered through the film was recorded at break time up to 25 h. The determinations were made in triplicate. 

### 2.9. Fourier-Transform Infrared Spectra of Edible Films

The infrared spectra of the films were recorded with an attenuated total reflectance-Fourier transform infrared (ATR-FT-IR) spectrometer (Spectrum Two, Perkin-Elmer, Waltham, MA, USA) covering wave numbers from 4000 to 600 cm^−1^ with a resolution of 4 cm^−1^, and 32 scans were averaged for each spectrum [22]. The thin films were applied directly onto the ATR cell. Before analysis, the films were kept in a desiccator with silica gel for 1 week so that the films were dehydrated. The measurements were made in triplicate.

### 2.10. Differential Calorimetric Analysis of Edible Films

The calorimetric analysis of the films was carried out as previously described (Arancibia, et al. [19]). An amount of 10 to 15 mg of films were weighed and encapsulated in aluminum containers. An empty capsule was used as a reference. The analysis of the samples was conducted under an inert nitrogen atmosphere (50 mL/min) in a differential calorimeter (DSC) DSC 3 (Mettler Toledo, Spain), with a temperature sweep from 30 to 200 °C with a heating rate of 10 °C/min. Data were normalized to dry matter content after drying each capsule. The measurements were made in triplicate.

### 2.11. Atomic Force Microscopy (AFM) Imaging

The surface morphology of the films was characterized using an atomic force microscopy (AFM) (Park Systems XE7, Santa Clara, CA, USA). The films were cut into thin pieces so that they could fit into the AFM imaging; they were stuck onto the sample stage with double-sided tape and scanned in noncontact mode. Analyses were developed in air conditions using a Nano sensor PPP-CONTSCR commercial silicon cantilever tip (0.2 N/m, 25 kHz, <7 nm typical radius). The software XEI permitted the linear flattening of the images, and the resulting data for each film was transformed into a 3D image [23]. No treatment was conducted on the sample for analysis. 

### 2.12. Microorganisms in the Edible Film

For the microorganism evaluation, the film with the best functional characteristics was selected. The selected edible film was tested for viable total aerobic mesophilic bacteria, *Enterobacteria*, and *Staphylococcus aureus*. The microbial load of the film after 1 day and 1 year of storage (58% RH and ambient temperature) were also tested in duplicate. An initial dilution was prepared using 1 g of film and 9 mL of 0.1% buffered peptone water (Difco, Sparks, MD, USA). Serial dilutions were prepared, and viable total aerobic mesophilic bacteria were tested using the pour plate technique with a plate count agar (PCA, Difco, Sparks, MD, USA) incubated at 30 °C for 72 h. *Enterobacteria* were tested using a double layer of Violet Red Bile Glucose Agar (VRBG, Difco, Sparks, MD, USA) incubated at 30 °C for 48 h. Presumptive *Staphylococcus aureus* was tested by spread plate onto Baird Parker agar (Difco, Sparks, MD, USA) with egg yolk tellurite emulsion (Merck Millipore, Temecula, CA, USA) incubated at 30 °C for 48 h. 

### 2.13. Antimicrobial Activity of the Film

The antimicrobial activity was evaluated in the film with the best functional characteristics using the disk diffusion method in agar plates [21]. The selected film was tested against several microorganisms to evaluate its antimicrobial properties. The film was cut into discs of 5 mm diameter and laid onto the surface of Brain Heart Infusion (BHI, Difco, Sparks, MD, USA) previously inoculated with some microorganisms selected for being responsible for food spoilage or for being potentially pathogenic organisms: *Enterococcus faecalis* ATCC 29212, *Salmonella enterica* ATCC 9842, *Escherichia coli* ATCC 25922 and *Listeria monocytogenes* ATCC 7644. Each determination was performed in duplicate. The clear zone (halo) surrounding the film discs was considered a measurement of antimicrobial activity. Results were expressed as the diameter of growth inhibition (mm).

### 2.14. Food Applications

For the food applications, the film with the best functional characteristics was selected. In this formulation, some powdered seasonings (0.2% cumin, 0.2% oregano, 1.4% garlic, 1.4% onion, 0.06% pepper, and 0.09% nutmeg) were incorporated and mixed into the film forming solution just before casting. The spices were added to confer some attractive flavor and/or simulate the culinary seasoning effect of commercial disposable films that impregnate flavors in some muscle-foods (although commercial films are discarded since they are non-edible). The use of the film as a protective wrap during cold storage was also evaluated.

#### 2.14.1. Antimicrobial Effectiveness of the Films on Food

Skinless, boneless chicken breasts (Pronaca, Ecuador) were acquired at a local market. Breasts were cut into fillets to obtain different batches (five fillets of 20 g for each batch) and treated as follows: packaged in a polyethylene bag, considered as the control (C), wrapped in an banana edible film incorporated with spices (SF), seasoned with spices in same quantity and ratio in which they were added in the production of the film (SC) (spices were pre-mixed and added on the breast) and wrapped in a plain banana edible film, without spices (CF). 

These batches were stored at ~5 °C for 7 days and periodically tested. For the determination of the antimicrobial effect of the films during storage, approximately 10 g of chicken breast were aseptically transferred into sterile Bags (Whirl-Pak, Madison, WI, USA) with 90 mL of 0.1% buffered peptone water (Difco, Sparks, MD, USA) and appropriate decimal dilutions in peptone water were prepared for the following microorganism determinations: (I) Total aerobic mesophiles on pour plates of PCA (Difco, Sparks, MD, USA)) incubated at 30 °C for 72 h. (II) *Enterobacteria* were tested using the pour plate method in a double layer of VRBG (Difco, Sparks, MD, USA) incubated at 30 °C for 48 h. (III) *Staphylococcus aureus* were tested by spread plate onto Baird Parker agar (Difco, Sparks, MD, USA) with egg yolk tellurite emulsion (Merck Millipore, Temecula, CA, USA) incubated at 30 °C for 48 h. All analyses were performed at days 0, 3, 5, and 7 in duplicate. The day when the chicken breast fillets were wrapped was considered day 0. Microbiological counts are expressed as the log of the colony-forming units per gram (log CFU/g) of the sample [17].

#### 2.14.2. Sensory Analysis of Edible Film 

Twelve semi-trained judges evaluated the sensory characteristics of the seasoned film in different presentations, (A) films without further heat treatment (SF), (B) with grilling treatment at 150 °C by 10 s side by side (SF-h), (C) as a wrap for chicken breast fillets (0.05 cm thick); once fillets were wrapped, the breasts were roasted 30 s per side in a frying pan previously heated at 150 °C (SF-w). In this treatment, judges were asked to evaluate the wrapped chicken as a whole (without detaching the film). The panelists evaluated color, texture, taste, odor, and general acceptability in the three different presentations. The panelists used a 5-point hedonic scale (5—very pleasant; 4—pleasant; 3—neither pleasant nor unpleasant; 2—unpleasant; 1—very unpleasant) for general acceptability, taste, appearance, and odor. A five attributes scale was used for texture (5—very hard; 4—hard; 3—neither hard nor soft; 2—soft; 1—very soft), for color (5—very dark; 4—slightly dark; 3—neither light nor dark; 2—light; 1—very light), and for crispness (very crispy, crispy, neither crispy nor soft, soft and very soft). The judges were also offered the possibility to provide comments or observations on each of the sample evaluations. The samples were evaluated in individual sessions. Samples were served to each panelist with water and crackers to cleanse the palate during the tasting.

### 2.15. Experimental Design

Results were analyzed by one-way ANOVA. Means were tested with Tukey’s Multiple Comparison Test, with a significance level α = 0.05 using GraphPad Prism v5.03 (GraphPad Software, San Diego, CA, USA).

## 3. Results and Discussion

### 3.1. Screening of Film Forming Ability

In order to select the most suitable formulation for the elaboration of the films by casting, some preliminary tests were carried out (Table 1). The addition of 0.5% whole banana flour, regardless of the percentage of banana peel flour and 1% plasticizer, did not allow the film to form. The formulation with an equal proportion of flours, whole banana, and peel flours (2:2), with 1% glycerol, partially improved the filmogenic capacity. This film detached from the surface with some difficulty, but it was manageable and less transparent. The concentration of whole banana flour was increased in order to enhance the properties of the film. So, using whole banana and peel flour (2.5 g: 1.5 g) with 1% glycerol, a proper film was obtained; this film was easily detached from the plate, manageable, steady, and presented a smooth surface. Increasing the plasticizer concentration up to 1.5% did not improve the properties; on the contrary, this percentage of glycerol modified the film structure forming a coarse film with a rough surface and turned opaque, although manageable and flexible. Thus, the selected films for further characterization were: whole banana flour (2.5%), peel flour (1.5%), with glycerol (1.0%), F-1.0 G film, and with 1.5% glycerol, F-1.5 G film.

### 3.2. Rheological Properties of Filmogenic Solutions 

Figure 1 shows the dynamic flow behavior of the film-forming solutions with different concentrations of glycerol. The elasticity modulus (G′) and viscosity modulus (G″) as a function of angular frequency (ω) were studied. The values of G′ and G″ for the F-1.0 G and F-1.5 G solutions increased (G′ > G″) with an increase in angular frequency. The slightly higher values of G′ and G″ were obtained in the sample with the lower glycerol concentration (F-1.0). Similar behaviors were found in starch films of different sources (aphia, cassava, and corn) containing glycerol compared with those that do not contain glycerol [24]. An increase in G’ values indicates that the solutions exhibit the typical characteristics of viscoelastic material [25,26]. It is important to note that G’ and G” displayed a classical gel behavior for the film-forming solution. This means that, at a given angular frequency, F-1.0 G and F-1.5 G would mainly display a weak gel behavior [27], probably related to the fiber content. Xiong, et al. [28] observed that incorporating carboxymethyl cellulose fibers in wheat starch reduced the rigidity and had a network-weakening role on amylose. On the other hand, the effect of glycerol on G’ and G” was significantly less pronounced. Similar results were observed by Chen, et al. [29] in edible films from tapioca starch/decolorized hsian-tsao leaf, where the effect of the plasticizer was minimal compared to the effect of fiber.

### 3.3. Characterization and Properties of Films

#### 3.3.1. Moisture, Aw, Thickness and Density

The moisture content was noticeably higher in F-1.5 G film than in F-1.0 G film (*p* < 0.05) (Table 2), which was due to the hygroscopic capacity of the plasticizer. In the present work, the moisture values of films (kept at 52% RH) were higher than those described in films based on 15.1% banana flour or 12.2% banana starch with much higher amounts of glycerol (19%) [30], kept at lower RH (48%). The moisture content of films could be associated with the large proportion of other hydrophilic components different from starch, such as proteins and fiber, which can increase the number of molecular interactions with water and induce a more porous structure that increases water retention by capillarity [11]. 

The water activity (aw) is an important parameter to evaluate the shelf-life of the films under different conditions. In this study, aw values were low and similar (*p* < 0.05) in F-1.0 G film and F-1.5 G film. Given the low values obtained, it is unlikely that the films suffer significant growth of microorganisms [31], providing high stability.

The film with the lower glycerol (F-1.0 G) content was less thick and dense (*p* < 0.05) than the F-1.5 G film (Table 2). The function of a plasticizer is to reduce the forces between polymeric chains by interposing between them, leaving free volume between the polymers [32]. These free volumes are then be filled with water which formation is favored with a higher glycerol content, hence their higher density in F-1.5 G film (despite the drying treatment in both films being similar). As for the density of both films, especially of F-1.5 G film, values were similar to those of other films made with starch or pulp banana flour (1.34 and 1.18 g/cm^3^, respectively) [30] (Table 2). Moore, et al. [33] found an increment of film density (0.92 to 1.10 g/cm^3^) with the increase in glycerol concentration in keratin films.

#### 3.3.2. Water Solubility

The water solubility of the film is an important property to assess its applicability in food and may also determine or favor its biodegradability. The water solubility of the F-1.5 G film was slightly higher than that of the F-1.0 G film due to the presence of a higher amount of glycerol (Table 2). These water solubility values were greater than those reported by Anchundia, et al. [34] in films based on banana peel flour (0.5–1.5%) and acetyl salicylic acid (1.5–2.5 mmol) as an adjuvant in preservation, probably due to the absence of traditional plasticizers in the films used by these authors. Generally, the solubility value of the films will depend on the assigned application or use; a low solubility value is necessary for the films when they are used as food coatings during storage and, on the contrary, a high solubility value is advantageous when cooking certain foods [35]. The solubility of films (F-1.0 G and F-1.5 G) is in the range of that obtained in a banana flour film (27.9%) and are also higher than film obtained with banana starch (21.3%) (both films formulation containing 19% glycerol) because the flour film has a more porous structure than the starch film [30]. These authors found that in the latter, the interactions between amylose, amylopectin, and amylose–amylopectin that occur during film drying reduce the amount of hydrophilic groups available to interact with water [30].

#### 3.3.3. Water Resistance

The water resistance of films expresses the ability of film to withstand water without breaking and the deformation produced by the weight of the water. After 48 h, both films showed a small deformation ≤0.5 cm due to the weight of the water on the film (Table 2). After 25 h, water dripping through both films was detected, the amount of water filtered did not exceed 1.75 mL. This water dripping caused a small pore (which was considered a breakage point) in F-1.0 G film, while in F-1.5 G film, no breakage or pores were appreciated, showing the formation of a stable matrix. The presence of more plasticizer and the tendency of the F-1.5 G film to hydrate did not negatively affect the water resistance. Blanco-Pascual, et al. [36] reported less resistance to water, a deformation of 1.5 cm after shorter times (8 h), and a breakage point at 32 min in a film made from extracts of algae (*Mastocarpus stellatus*) and carrageenan. The presence of protein material favored the formation of interactions and reinforcement of the filmogenic network, increasing the resistance to water [37].

#### 3.3.4. Water Vapor Permeability

The water vapor permeability (WVP) of F-1.0 G film was higher than F-1.5 G film (*p* < 0.05) (Table 2). The WVP values were considerably lower than those reported for a film made with different flours: achira flour (4% *w*/*w*) with 17 g glycerol/100 g flour (45.79 ± 0.2 g mm m^−2^ d^−1^ KPa^−1^), amaranth flour (4% *w*/*w*) with 0.9 g/100 g solution (6.05 ± 0.2 g mm m^−2^ d^−1^ KPa^−1^), banana flour (4% *w*/*w*) with 19 g glycerol/100 g flour (18.14 ± 0.2 g mm m^−2^ d^−1^ KPa^−1^) [30]. Accordingly, a higher WVP has been reported in plasticized (glycerol, sorbitol, and a mix of glycerol-sorbitol) banana-based films [38]. In films based on mixtures of carboxymethyl cellulose (CMC) or gelatin with concentrations of 4% banana flour and 1.2% glycerol, the WVP decreased with increasing CMC/gelatin concentration, being higher (14.90 to 20.76 g mm m^−2^ d^−1^ KPa^−1^) [39] than those reported in the present study. The low WVP in F-1.0 G and F-1.5 G films could be favored by a low percentage of plasticizer and a high fiber content [40] (Table 2).

#### 3.3.5. Color and Light Barrier Properties of Edible Film

The color measurement based on L*, a*, and b* parameters (Table 2) is shown. The F-1.0 G film was lighter in color and slightly more transparent than the F-1.5 G film. These results confirm the preliminary data obtained by screening (Table 1). The higher glycerol content retains a higher amount of water, thus constituting a thicker and denser film, which is consequently less translucent. 

Lightness was low in both films but even lower in F-1.5 G film (*p* < 0.05). The redness (a*) and yellowness (b*) values, as was expected, showed slight differences (*p* < 0.05) since the formulations were very similar. Considering the parameters L* and b*, the films are darker with a tendency to yellowish tones, which reflect the color of raw materials. Traditionally, the yellow color in the fruits indicates the presence of flavonoids, and intense red colorations are attributed to carotenoids and anthocyanins, among others [41]. Pelissari, et al. [30] reported negative values close to zero for a*, indicating the absence of red color and yellowish tendency in flours from banana. 

Transparency in visible and ultraviolet wavelengths was low in both films, even lower in F-1.5 G film. The low transmittance of light in the visible and UV spectrum is important since light is an initiator of oxidation in foods, which can be subjected to both types of light in supermarket displays for long periods. Opacity could be associated with the content of proteins, fiber, and phenolic compounds found in peel flour. The amylose content influences the opacity of the films, in which linear molecules favor hydrogen bonds between the hydroxyl groups of neighboring chains, decreasing their interactions with water [30], resulting in an opaque polymer matrix [42], and may have taken place to a greater extent in F-1.5 G film. In banana flour films with 19% glycerol. Pelissari, et al. [30] reported lower opacity values (51.3%) than those found in F-1.0 G film (75.6%) and F-1.5 G film (87.6%). The presence of the plasticizer used by the authors mentioned above may be the cause of this lower opacity, as it could facilitate a hollowing of the matrix and favor transparency; in addition, they did not use the banana peel, so the fiber content was probably lower, and the color of the flour was not so dark. 

### 3.4. Mechanical Properties 

F-1.5 G film registered higher puncture force and deformation values than F-1.0 G film (Table 2). Both films showed greater puncture (8.15 N for F-1.0 G and 1.20 N for F-1.5 G) and deformation (7.74% for F-1.0 G and 9.92 for F-1.5 G) test values than those in pulp banana flour (6.0 N and 6.8%, respectively) and starch (8.1 N and 1.3%, respectively) films reported by [30]. This fact can be possibly due to the high-fiber content of the films in the present work (made of whole banana and banana peel flours), which improves the mechanical properties because of the strong interactions between fiber and the starch matrix allowing a better stress distribution [32]. However, the tensile strength, similar in both films (*p* < 0.05), was lower than the 7.97 MPa presented by Anchundia, et al. [34] in films produced with 1% banana peel (*Musa paradisiaca*) and 2 mmol/L of acetyl salicylic acid, and also in films from banana (*Musa paradisiaca*) flour with 19% glycerol (6.8 MPa) [30], perhaps as a result of the type and quantity of the flour [43]. Likewise, tensile strength values are higher than those reported in edible coatings from ripe “Prata” banana peel flour (0.14–0.70 MPa) [44]. The elongation at break was higher in F-1.5 G film (*p* < 0.05), which is related to glycerol presence, which increases the flexibility and elongation of the films. The modulus of elasticity was low and similar in both films (*p* > 0.05), possibly due to proteins, lipids, and water and their interactions that contribute to the plasticizing effect, favoring flexibility and showing that proteins do not contribute to forming a stronger matrix [30].

### 3.5. Fourier-Transform Infrared (FTIR) Spectroscopy

In the IR spectrum of the films (Figure 2), most bands corresponded to carbohydrates and fibers, and there were very few differences as expected in such similar formulations. The first dominant band from 3000 to 3600 cm^−1^ can be assigned to the extension of free and molecularly bound hydroxyl groups present in carbohydrates. Bands at 2930 and 2885 cm^−1^ were also observed, indicating extensions of the CH2 groups, while the bands located between 2800 and 3000 cm^−1^ could be attributed to the amylose and amylopectin structures lignin from the peel or lipids [45]. A large number of adsorption bands were observed in the region from 1700 to 1200 cm^−1^, probably from minor flour components such as proteins and lipids. The band at 1637 cm^−1^ which corresponds to the stretching of the C=O bond (C=O stretching) from the Amide I, could be highly affected by crystallinity variations in starch and was slightly higher in the F-1.0 G film compared with the F-1.5 G film. This might be due to the water absorbed in the amorphous region of starch [30,46]. A band at 1417 cm^−1^ associated with the symmetric extension of the carboxyl group (–COO) was also observed. In addition, the band at 1242 cm^−1^ corresponds to C-O stretching from the lignin [47]. Indeed, this peak could be assigned to the cellulose stretching in the 1400 cm^−1^ region and was more intense in the F-1.5 G film. Tibolla, et al. [48] attribute it to a higher fiber content. The band at 1150 cm^−1^ in the F-1.0 G film appeared slightly shifted to a higher wavelength in F-1.5 G film, probably due to the breaking of hydrogen bonding with water molecules by the glycerol content. Hydrogen bond interactions could decrease due to the low concentration of plasticizer [49], as was observed in the F-1.0 G film. Bands at 1078, 1104, and 1150 cm^−1^ from both spectra are commonly attributed to the stretching of C–C, C–O, and bonds C–O–H of anhydro-glucose ring commonly found in the structure of carbohydrates [50,51]. The band at 926 cm^−1^ is associated with glycosidic bonds in starch and the presence of amylopectin α- bonds. The band at 670 cm^−1^ could be attributed to the phenolic compounds that could be slightly free or exposed in F-1.5 G film in contrast to F-1.0 G film (Figure 2). 

### 3.6. Differential Scanning Calorimetry (DSC) 

The banana flour films exhibited a single sharp endothermic peak (Figure 3), corresponding to the melting temperatures (Tm) 122.39 ± 4.26 °C and 122.72 ± 0.48 °C of F-1.0 G and F-1.5 G films, respectively. The presence of a single peak indicates homogeneity and could be related to the melting of crystalline starch domains [52]. In general, adding plasticizers decreases the thermophysical transition temperature by the disruption of crystals [53,54]. Nevertheless, no significant differences were observed between both films (*p* < 0.05). The effects of plasticizers such as glycerol on thermal properties are difficult to evaluate due to the interaction of water and other plasticizers in the formation of the film [55]. However, F-1.5 G film required more enthalpy (ΔH: 250.46 ± 19.32 J/g) than F-1.0 G film (ΔH: 187.61 ± 28.90) (data obtained from thermogram) probably because the greater presence of plasticizer (glycerol and water) hinders this thermal transition. Higher enthalpy values were observed in film blends of cassava starch and CMC, with increasing CMC [52] or when increasing the content of cellulose in banana peel films [56].

### 3.7. Atomic Force Microscopy (AFM) Imaging

The images obtained by atomic force microscopy (AFM) show the topographic nanoscale characterization of the surface of the films produced with whole banana and banana peel flours (Figure 4). This structural analysis allows for a possible explanation for the elastic results obtained at the macroscale, such as porosity, permeability, flexibility, or mechanical resistance [57,58]. F-1.0 G and F-1.5 G films showed higher roughness, (Sq: root-mean-square) in the first case: Sq (F-1.0 G film) = 30.1 nm, Sq (F-1.5 G film) = 11.1 nm, for a scanned surface of 20 µm × 20 µm. One possible explanation involves a decrease in the size of the colloids formed in the F-1.5 G film due to the greater presence of glycerol in the mixture compared with the F-1.0 G film. Moreover, it should be noted that both types of films contain the same amount of flour, so the improvement in dilution homogenization could be mainly attributed to an increase in the plasticizing effect created by glycerol [59]. Additionally, it is noted that an increase in the roughness of the film is characterized by a non-compact network which increases the water vapor permeation [57]. In the present work, the vapor permeation of F-1.0 G film is slightly higher than that of F-1.5 G film; similarly, the former was less dense (Table 2) creating a more porous structure.

### 3.8. Microorganisms in the Edible Film

The microbial analyses showed that the film was negative for the target microorganisms *Enterobacteria* and *Staphylococcus aureus*, which could be related to the low aw (0.62) in the edible film. Counts for total aerobic mesophilic bacteria (freshly prepared—after 1 day-, and after 1 year of storage) was around 10^4^ CFU/g (data not shown). These results provide a criterion of the film quality and stability throughout storage under controlled conditions. Perhaps these microorganisms are acquired during the conditioning process since the procedure is not performed under sterile conditions. The casting technique involves a drying process that could protect the films by helping to reduce the viability of microorganisms [60]. This fact, together with the low water activity of the film, makes the development of viable microorganisms unlikely. These results cannot be contrasted with the literature because there are no reports on the microflora of edible films from whole banana flour. 

### 3.9. Antimicrobial Activity

The antimicrobial activity of the F-1.0 G film was also tested, and they showed no antimicrobial properties (data not shown) against the microorganisms studied (*E. faecalis* ATCC 29212, *S. enterica* ATCC 9842, *E. coli* ATCC 25922, *L. monocytogenes* ATCC 7644). This fact is probably due to a low concentration of antimicrobial agents such as polyphenols reported in this type of cultivars [61].

### 3.10. Food Applications 

As previously mentioned, the film with the best filmogenic characteristics (F-1.0 G film) was selected for food applications. The film can have different purposes: as a snack per se; as a separator when the food product is presented in slices, small portions, etc., preventing them from sticking together; as a wrap; as a substitute for single-use plastics that are usually in contact with the food, etc. In addition, these films were heat-sealable. The ability to preserve food and the behavior of the film as a food wrap the following tests were performed. 

#### 3.10.1. Antimicrobial Effectiveness of the Films on Food

To evaluate the effect of this film as a primary packaging of food (wrap) and its potential as a substitute for single-use plastics, a study in refrigeration was carried out, for which chicken breast was chosen. 

Four batches of chicken breast were studied: chicken breast packaged in a polyethylene bag (C), seasoned with spices (SC), wrapped in a banana edible film-without spices (FC), and wrapped in a banana edible film-incorporated with spices (SF). The initial total aerobic mesophile count in the raw chicken breast was around 4.5 log (CFU/g) (Figure 5A). Similarly, Takma, et al. [62] reported counts of around 4.53 log (CFU/g) in chicken breast meat stored at 4 °C. All the batches exhibited similar behavior as the counts increased over time. C and FC exhibited the fastest increase in microbial counts. The use of spices and their incorporation into the film allowed to obtain 1 log cycle counts lower than the control and the film without spices over 7 days of storage, reaching around 4.8 log CFU/g at the end of the trial (*p* ≤ 0.05). So, the increment in total mesophilic counts in SF and SC was barely noticeable after 7 days of storage. These results may have been due to the antimicrobial effect of the spices, including garlic. 

An increment of microorganism after 7 days of storage was observed in counts in Bard-Parker (presumptive *Staphylococcus aureus*) and *Enterobacteria* (Figure 5B,C respectively). The initial count of *S. aureus* in fresh chicken breast meat was 1.5 log CFU/g which agreed with the values observed by Mohammadi, et al. [63] in fresh chicken meat during refrigerated storage (2.75 log CFU/g). At the end of the storage, the SC and SF samples showed lower values than C and FC (*p* ≤ 0.05) for the evaluated microorganism, showing an evident effect on *S. aureus*. In an in vitro study, Chikwem, et al. [64] reported a significant antimicrobial activity from garlic against S. aureus, which, together with the rest of the spices added and whose antimicrobial activity has been previously described [63,64,65,66,67,68,69], could explain the obtained results. 

The initial count of *Enterobacteria* was 1.8 log CFU/g, lower values than those observed by Raeisi, et al. [70] in fresh chicken meat. During the whole storage, the *Enterobacteria* count gradually increased (Figure 5C). Batch SF and SC attained the lowest number in total *Enterobacteria* compared to the control and FC batch (*p* ≤ 0.05). SF and SC were most effective in controlling the microbial growth throughout the storage time, probably due to synergistic antimicrobial effects of the ingredients used to spice up the chicken or film. In addition to garlic, spices such as cumin, oregano, garlic, onion, black pepper, and nutmeg have been reported to be potent antimicrobial agents [67,71]. Overall, the use of spices incorporated into the edible film could have enhanced the antimicrobial effect, inhibiting the growth of mesophiles, *Enterobacteria*, and *S. aureus* and making the chicken breast more palatable and safer for consumption. Although the film without spices did not show any antimicrobial activity, it showed slightly lower counts than C, perhaps due to limiting the aerobic conditions to a certain extent by wrapping the chicken with the film. The same fact is observed when comparing the effect of incorporating spices directly vs. in seasoned films; although the difference in counts is small, the chicken retains moisture and has a better appearance when it is wrapped and on the grill, which will have a positive impact during its culinary treatment and sensory evaluation.

#### 3.10.2. Grill-Toasted Edible Film

The chicken breast wrapped in the seasoned film was toasted on the grill pan (Figure 6A,B). Once the heat treatment finished, the film was easily separated, observing that the breasts had acquired a ready-to-eat, toasted appearance (Figure 6C). The film is completely edible (Figure 6B), so whether it is for taste, desire and/or confusion, the film can be eaten with the chicken because of the natural ingredients in its formulation. Integrity, malleability, and wrapping behavior were evaluated before and after cooking, while during cooking, integrity, burning, and strange or undesirable odors were also evaluated (data not shown). None of these appeared during cooking.

This wrap is intended to be an analog or substitute for various existing products in the market, such as paper or “ready to pan” bags, both non-edible, used to season food, mainly meat, fish, poultry, or vegetables. In the present work, in addition to adding spices, the seasoned film protected the food, chicken in this case, from the heat source, making the chicken juicier and more tender, and allowing cooking without using breadcrumbs and any fat, although the appearance of the film was reminiscent of a conventional breading, or even a tortilla used to make stuffed patties.

The film as it can be consumed alone in two different ways: (i) without heat treatment (SF), as a product reminiscent of nori seaweed (Figure 6D); (ii) with heat treatment (roasted) (SF-h) conceived as a “nacho-chip” type product or similar (Figure 6E).

#### 3.10.3. Sensory Evaluation of the Edible Film 

The results of the sensory scores of the spiced film as chicken breast wrap (SF-w), spiced film without additional heat treatment (SF), and spiced film with heat treatment (SF-h) are shown in Figure 7. 

In the wrapped chicken (SF-w) (Figure 7A), the texture was well qualified: the chicken presented a good palatable texture, and the judges also mentioned that it was similar to fried chicken with its own skin, with the advantage of the absence of fat, showing excellent acceptability. Additionally, it was observed that the wrap was also crispy. 

SF-h and SF (Figure 7B) also showed good acceptability. In the texture parameter, the judges classified SF as “neither hard nor soft”; however, when SF-h was evaluated, the score was close to very hard and very crispy. This qualification is important because it is reminiscent of a typical “nacho-chip” cereal-type snack.

Regarding taste, appearance, and odor, all samples were considered “pleasant” indicating that the judges accepted well the films in the different presentations. The color of the three films (SF, SF-h, and SF-w) was categorized as slightly dark because of the presence of proteins and fiber from the banana flour and the spices used in the formulation.

## 4. Conclusions

The mixture of whole banana flour and banana peel flour with glycerol showed good filmogenic capacity. The two selected formulas presented very slight differences, but this resulted in the greater water uptake of F-1.5 G film due to its higher hygroscopicity power, which led to an increase in film density, thus compacting it and making it less rough. However, in the film with less glycerol (F-1.0 G), the structure was rougher and slightly less elastic, deformable, and firm. The film (F-1.0 G) selected for further food applications was stable, maintaining the microbial load practically constant after one year of storage. This film did not show any antimicrobial activity. During the chicken storage trial, the spice-free film showed an effect similar to that of conventional plastic. When wrapping the chicken with a spiced film, the reduction of at least one log cycle was observed for total mesophilic aerobic microorganisms, *Enterobacteria* and *Staphylococcus aureus*, a reduction which in general was higher than in the spiced chicken alone.

The sensory evaluation allowed us to establish good general acceptability in the three evaluated presentations, as two different types of snacks and as a chicken wrap. The use of flours from discarded green bananas (from the whole banana and the banana peel) in the development of an edible packaging for food applications could be a good strategy to take advantage of an underused resource that contains a good source of nutritional components while avoiding the use of plastic or other non-environmentally friendly packaging.

## Figures and Tables

**Figure 1 polymers-13-03183-f001:**
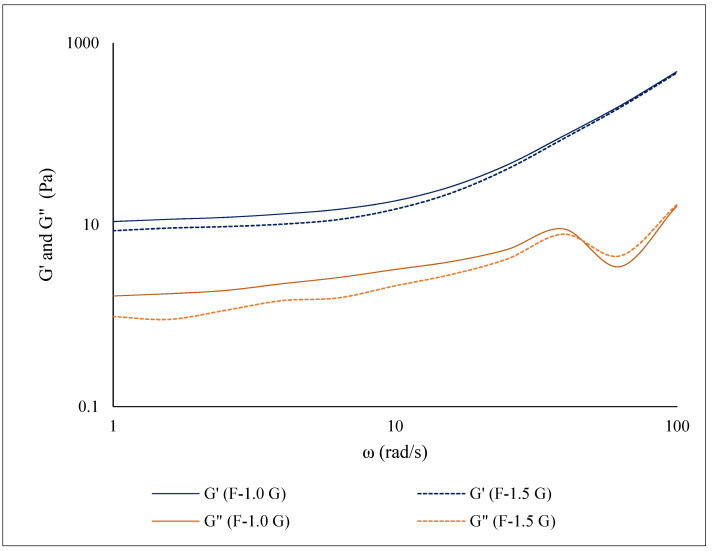
Rheological behavior of films formulated with green banana flours.

**Figure 2 polymers-13-03183-f002:**
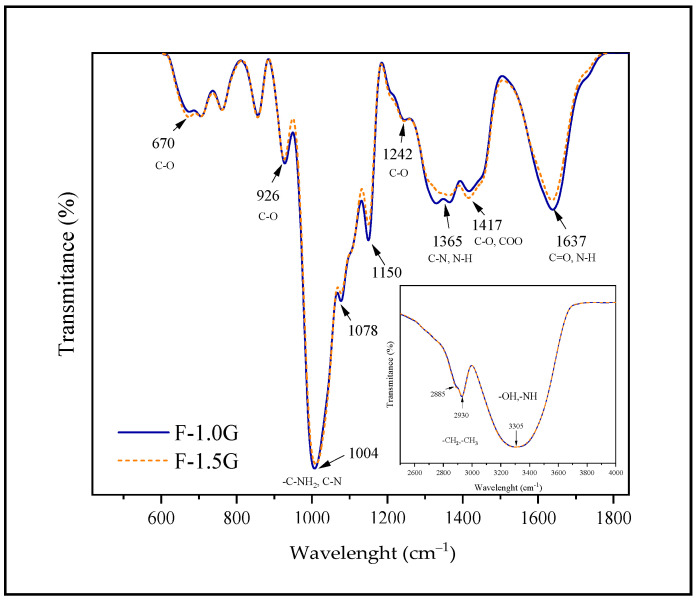
Fourier-transform infrared (FTIR) spectroscopy of the films formulated with green banana flours.

**Figure 3 polymers-13-03183-f003:**
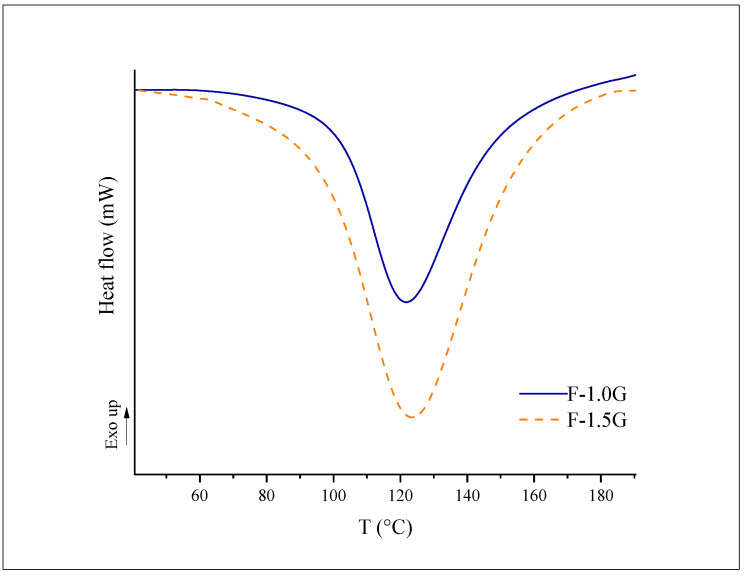
DSC thermogram from films formulated with green banana flours.

**Figure 4 polymers-13-03183-f004:**
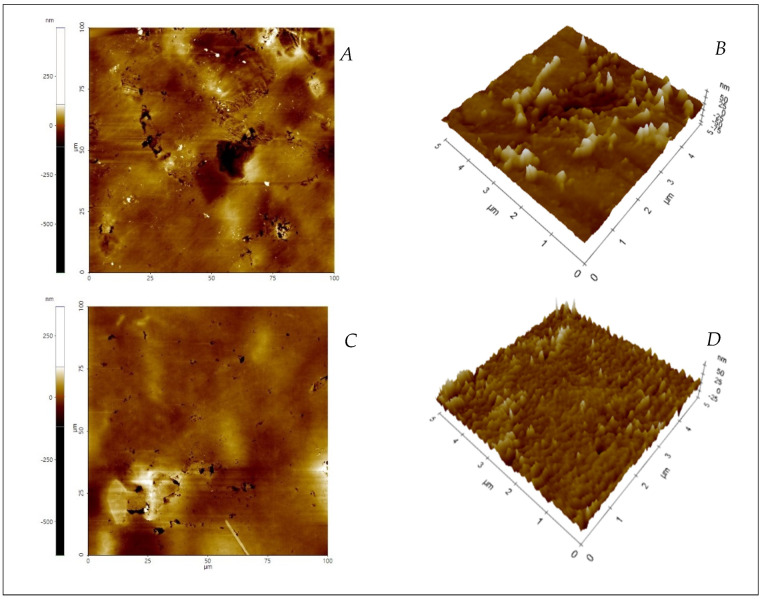
Topographic AFM contact mode images from films formulated with green banana flours. (**A**) 2D image of F-1.0 G film; (**B**) 3D image of F-1.0 G film; (**C**) 2D image of F-1.5 G film; (**D**) 3D Image of F-1.5 G film.

**Figure 5 polymers-13-03183-f005:**
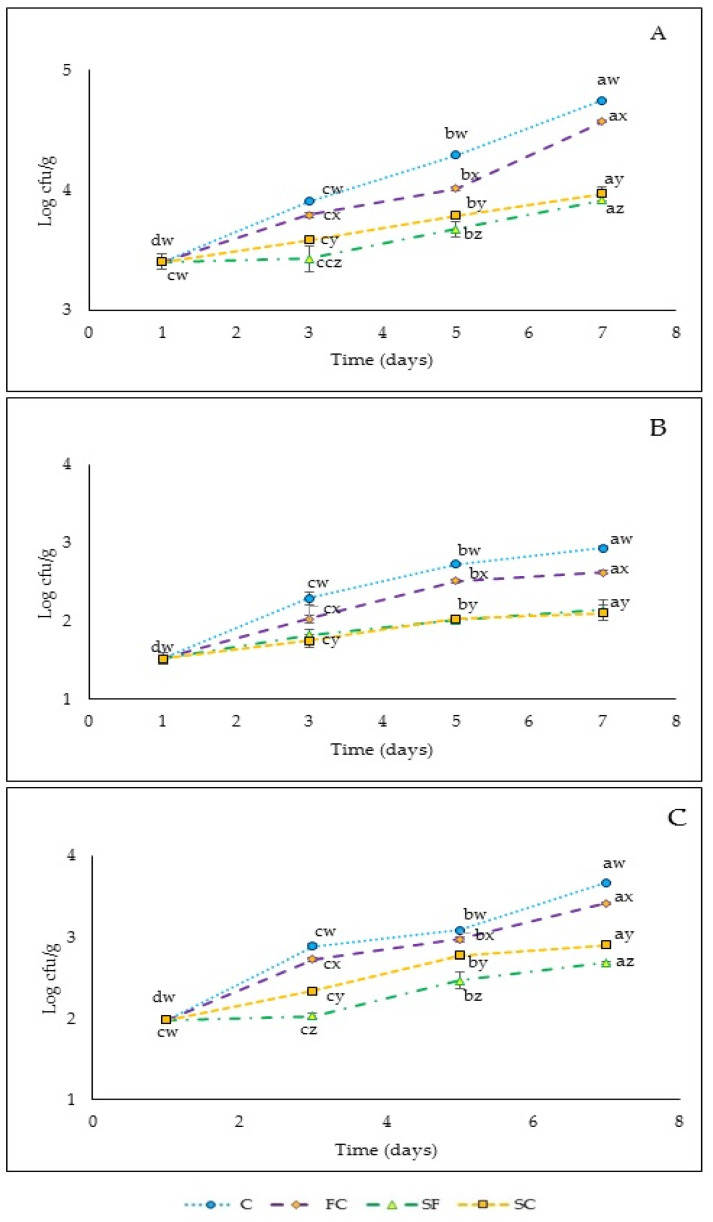
(**A**) Total aerobic mesophiles bacteria (cfu/g), (**B**) *Staphylococcus aureus* (cfu/g), and (**C**) *Enterobacteria* (cfu/g), during chilled storage of boneless chicken breast packaged in a polyethylene bag (C), seasoned with spices in same ratio as was added for film production (SC), wrapped in a banana edible film without spices (FC), and wrapped in a banana edible film incorporated with spices (SF). Values are the mean ± standard deviation. One-way ANOVA: different letters (a, b, c) in the same line show significant differences among days for each batch (*p* < 0.05). Different letters (x, y, z) in the same days show significant differences among batches for each day (*p* < 0.05).

**Figure 6 polymers-13-03183-f006:**
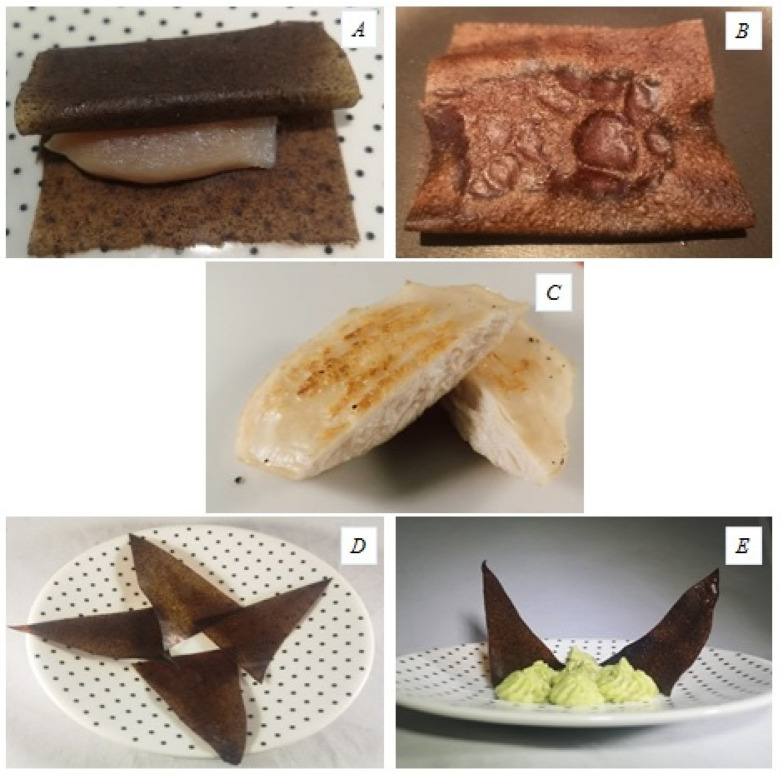
(**A**) Chicken breast wrapped in seasoned film, (**B**) Chicken breast wrapped toasted in a pan, (**C**) Chicken breast wrapped toasted opened after grilling, (**D**) Film without heat treatment (SF), and (**E**) Film toasted (SF-h) both as a snack.

**Figure 7 polymers-13-03183-f007:**
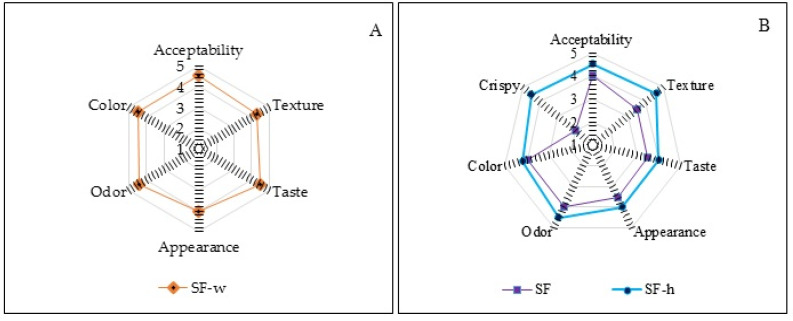
Sensorial evaluation of: (**A**) Wrap on chicken breast (SF-w). (**B**) Film with grilling treatment (SF-h) and spiced film (SF).

**Table 1 polymers-13-03183-t001:** Filmogenic ability, peeling off, handle, and transparency response of preliminary formulation.

PF (%)	WF (%)	G (%)	Film Forming Ability	Peeling Off	Handle	Transparency
0.5	0.5	1	_	_	_	++
1.5	0.5	1	_	_	_	++
2.5	0.5	1	_	_	_	++
3.5	0.5	1	_	_	_	++
2	2	1	+	+	++	_
2.5	1.5	1	+	+	++	_
1.5	2.5	1	++	++	++	+
1.5	2.5	1.5	++	++	+	-

PF: peel banana flour; WF: whole banana flour; G: glycerol.

**Table 2 polymers-13-03183-t002:** Properties of edible film produced with banana flours blends.

Properties	F-1.0 G	F-1.5 G
Moisture (%)	19.33 ± 0.92 ^b^	26.59 ± 2.59 ^a^
Water activity (aw)	0.62 ± 0.003 ^a^	0.62 ± 0.003 ^a^
Thickness (μm)	110.91 ± 0.14 ^b^	152.40 ± 0.79 ^a^
Density (g/cm^3^)	1.20 ± 0.02 ^b^	1.45 ± 0.02 ^a^
WVP (g mm m^−2^ d^−1^ KPa^−1^)	3.63 ± 0.12 ^a^	3.45 ± 0.12 ^b^
Water solubility (%)	24.9 ± 0.10 ^b^	27.38 ± 0.11 ^a^
Deformation at WR * at 48 h (cm)	0.50 ± 0.10 ^a^	0.45 ± 0.03 ^a^
Filtered volume at WR (mL)	3.37 ± 0.14 ^a^	3.25 ± 0.14 ^a^
L*	23.92 ± 0.14 ^a^	19.94 ± 0.45 ^b^
a*	4.73 ± 0.01 ^b^	5.77 ± 0.04 ^a^
b*	12.28 ± 0.03 ^a^	11.10 ± 0.19 ^b^
Transparency at 280 nm (%)	24.34 ± 0.22 ^a^	12.37 ± 0.91 ^b^
Transparency at 660 nm (%)	38.25 ± 0.23 ^a^	19.22 ± 0.19 ^b^
Puncture force (N)	8.15 ± 0.19 ^b^	11.20 ± 0.11 ^a^
Puncture deformation (%)	7.74 ± 0.28 ^b^	9.92 ± 0.26 ^a^
Tensile strength (MPa)	4.64 ± 0.48 ^a^	4.35 ± 0.14 ^a^
Elongation at break (%)	16.78 ± 0.20 ^b^	18.84 ± 0.19 ^a^
Modulus of elasticity (MPa)	37.58 ± 0.2 ^a^	38.14 ± 0.41 ^a^

F-1.0 G: film with 1.0% glycerol; F-1.5 G: film with 1.5% glycerol. Different letters (^a^, ^b^) indicate significant differences between samples (*p* < 0.05). * WR= water resistance.

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
