# Peer review of "Green Banana (*Musa acuminata AAA*) Wastes to Develop an Edible Film for Food Applications"

_polymers, 2021, doi:10.3390/polym13183183_

Round 1
Reviewer 1 Report
In this paper, the authors reported an edible film prepared with discarded green banana (Musa acuminata AAA) flour (whole banana and banana peel flours). The paper fit the aims and scope of Polymers. I would recommend accepting the paper after modifications. I have some comments to the authors.
- Introduction should be seriously improved. The novelty of the manuscript should be clarified in Introduction. In my opinion, source of raw materials which can be obtained from Agro-food wastes, is worth to be emphasized. The reuse of waste or by-products can increase economic value and environmental benefits and better highlight sustainability. Other bioactive substances based on natural ingredients from food byproducts capable of extending the food shelf life in a safe manner, especially antibacterial materials, could be used as examples..
doi: 10.1016/j.lwt.2021.111617, doi 10.1021/acs.jafc.0c00945, doi:10.3390/foods8080286
- The development of edible packaging materials should be introduced. Some very recent literatures should be mentioned.
doi:10.3390/foods9040449, doi:10.1016/j.foodhyd.2018.11.051, doi:10.1016/j.jfoodeng.2021.110697
- It is strongly suggested to indicate at the end of the Introduction section the main employed characterisation techniques in order to achieve their purpose.
- The method of FTIR should be described in details. How about the resolution? Why chose the range of 4000-600 cm-1
- The author should describe the method in detail instead of asking the reader to consult the literature.(Section 2.4)
- Section 3 should be changed to Results and discussion.
- It was recommended to evaluated the antimicrobial activity quantificationally.
Author Response
"Please see the attachment."

Reviewer 2 Report
The manuscript is interesting and the results well discussed. However some minor revision must be applied. The comment are reported in the attached pdf file

Author Response
"Please see the attachment."
